# Circulating microRNAs as Promising Biomarkers in Colorectal Cancer

**DOI:** 10.3390/cancers11070898

**Published:** 2019-06-27

**Authors:** Óscar Rapado-González, Ana Álvarez-Castro, Rafael López-López, José Iglesias-Canle, María Mercedes Suárez-Cunqueiro, Laura Muinelo-Romay

**Affiliations:** 1Department of Surgery and Medical Surgical Specialties, Medicine and Dentistry School, University of Santiago de Compostela, 15782 Santiago de Compostela, Spain; 2Liquid Biopsy Analysis Unit, Translational Medical Oncology (Oncomet), Health Research Institute of Santiago (IDIS), 15706 Santiago de Compostela, Spain; 3Instituto de Salud Carlos III, Centro de Investigación Biomédica en Red de Cáncer (CIBERONC), 28029 Madrid, Spain; 4Medical Digestive Service, Complexo Hospitalario Universitario de Santiago de Compostela (SERGAS), 15706 Santiago de Compostela, Spain; 5Translational Medical Oncology (Oncomet), Health Research Foundation Institute of Santiago (IDIS), Complexo Hospitalario Universitario de Santiago de Compostela (SERGAS), 15706 Santiago de Compostela, Spain; 6Oral Sciences Research Group, Health Research Institute of Santiago (IDIS), 15706 Santiago de Compostela, Spain

**Keywords:** colorectal cancer, circulating microRNAs, circulating exosomes, tumor biomarkers, liquid biopsy, personalized therapy

## Abstract

Colorectal cancer (CRC) is one of the most common cancers and a leading cause of cancer-related deaths worldwide. Despite numerous advances in therapeutic approaches, this cancer has a poor prognosis when it is diagnosed at late stages. Therefore, the scientific effort is nowadays directed towards the development of new non-invasive and dynamic biomarkers to improve the survival expectancy of CRC patients. In this sense, deregulated expression of many miRNAs has been shown to play an important role for CRC carcinogenesis and dissemination. Noticeably, an increasing number of studies highlight that circulating miRNAs, including those traveling inside exosomes or those released by tumor cells into circulation, constitute a promising tool for early detection, prognosis and therapy selection of CRC. Therefore, in this review we focus on the clinical potential of blood circulating miRNAs as emerging biomarkers with high value to improve the clinical management of CRC patients, providing a deep and complete perspective of the realities and challenges to translate these biomarkers to the clinical context.

## 1. Introduction

Colorectal cancer (CRC) is the third most commonly occurring cancer in men and the second among women and represents the fourth leading cause of cancer-related deaths worldwide [1]. Although CRC incidence and mortality rates vary widely worldwide, its burden is expected to increase by 60% to more than 2.2 million new cases and 1.1 million cancer deaths by 2030 [2]. The risk of CRC development is associated with numerous factors, including age, family history and inflammatory bowel diseases, diet or lifestyle, and only a small number of cases are due to underlying genetic disorders [3]. Hereditary CRC syndromes such as Lynch syndrome, familial adenomatous polyposis, MUTYH-associated polyposis, Peutz-Jeghers syndrome, juvenile polyposis and Cowden/PTEN hamartoma syndrome represent only between 2% and 5% of all CRCs [4]. Colorectal carcinogenesis is associated to three important genetic alterations (chromosomal, microsatellite and epigenetic instabilities) which play a key role in the two morphologic multistep pathways: the traditional model adenoma-carcinoma and the serrated neoplasia [5,6]. Importantly, 60–70% of diagnosed cases in non-screened symptomatic patients are detected in the advanced stage of the disease [7].

Nowadays, several CRC screening modalities are available ranging from non-invasive to invasive tests characterized by specific advantages and disadvantages. Fecal occult blood test and fecal immunochemical test have low sensitivity for detecting pre-neoplastic lesions and a high rate of false positives, while colonoscopy and flexible sigmoidoscopy, which are invasive diagnostic techniques, might be painful and are associated with low adherence rates [8]. To improve CRC, screening different blood-based biomarkers has been investigated as a potential diagnostic tool. Carcinoembryonic antigen (CEA) is a traditional blood-based biomarker widely used for monitoring CRC recurrence but does not represent a suitable diagnostic biomarker due to its low sensitivity and specificity [9]. Importantly, methylated Septin9 assay was approved by the Food and Drug Administration as the first non-invasive blood-based test for CRC screening, however, it has a limited sensitivity to detect adenomas or polyps [10]. Therefore, it is necessary to identify new non-invasive biomarkers with high sensitivity and specificity that could help in early CRC diagnosis, prognosis and treatment.

MicroRNAs (miRNAs or miRs) are a class of small (18–22 nucleotide) non-coding RNAs derived from hairpin precursors that post-transcriptionally regulate gene expression through targeting mRNAs [11]. MiRNAs are known to be involved in a wide variety of biological and pathological process including cancer, having an important role in cell proliferation, migration, invasion and metastasis [12]. In addition, miRNAs can regulate the tumor microenvironment, acting on tumor angiogenesis, immune system and tumor–stromal interactions [12,13]. It is also widely accepted that upregulation of miRNAs entails oncogenesis by downregulating tumor suppressor genes [14]. They modulate specific individual mRNA targets or multiple mRNAs [15] and show aberrant expression patterns in different tumor types [16,17,18].

Since the discovery of circulating miRNAs, numerous researchers have highlighted their potential as specific biomarkers for cancer. MiRNAs have been detected in various biological fluids including serum, plasma, urine, tears, saliva, seminal fluid, cerebrospinal fluid, extracellular fluid, and others [19]. Circulating miRNAs show a remarkable stability because they are protected against endogenous RNAses by different protective mechanisms such as miRNA modifications, miRNAs encapsulation by membranous extracellular vesicles or miRNA-binding proteins as Argonaute 2 (Ago2) or high-density lipoproteins [20]. Three major pathways could explain the presence of miRNAs in body fluids: (1) passive leakage of cellular miRNAs into the circulation due to cell apoptosis and tissue damage, (2) active selective secretion as microvesicles-free miRNAs (cell-free miRNAs) and (3) active secretion via binding to cell-derived microvesicles and exosomes [21]. Therefore, the expression profile of circulating miRNAs may reflect the miRNA profile of primary tumor or metastatic lesions, facilitating the detection of the tumor at different stages in a non-invasive manner. This approach represents a unique opportunity to discover novel biomarkers involved in CRC carcinogenesis. Considering the potential of miRNAs as valuable CRC biomarkers, we provide an overview of scientific evidence of circulating miRNAs for diagnosis, prognosis and therapy prediction in CRC patients (Figure 1 and Appendix A).

## 2. MiRNAs as Diagnostic Biomarkers in CRC

The first report that assessed the use of circulating miRNAs as a non-invasive diagnostic test for CRC was performed in 2009 [22]. Since then, several researchers have focused on the diagnostic value of circulating miRNAs for CRC detection [23,24]. Furthermore, ongoing research is aimed to identify specific miRNA profiles for early and advanced CRC stages, and even determine if there is a correlation with the tumor-node-metastasis (TNM) staging system. The discovery of miRNA signature involved in each step of the disease will increase the diagnostic accuracy in CRC. Therefore, circulating miRNAs could be potential non-invasive biomarkers for CRC diagnosis (Figure 2 and Appendix A).

### 2.1. Cell-Free miRNAs for Early CRC Diagnosis

Detection of precancerous lesions and early-stage CRC (I–II) is essential to reduce mortality. In the last few years, several studies [25,26,27,28] have evaluated the potential of circulating miRNAs as diagnostic biomarkers in CRC. Huang et al. were the first to evaluate the diagnostic value of plasma miRNAs for early CRC detection. They found significant elevated expression of miR-29a and miR-92a in 37 patients with advanced adenomas in comparison to healthy controls. Receiver operating characteristic (ROC) curve analysis yielded an area under the curve (AUC) of 0.769 for miR-29a with 62.2% sensitivity and 84.7% specificity, and an AUC of 0.749 for miR-92a with 64.9% sensitivity and 81.4% specificity. However, when both miRNAs were combined, an increase of the AUC value was observed [25]. Further research has revealed different miRNAs as potential biomarkers for early CRC diagnosis in serum and plasma. Wang et al. profiled 86 differentially expressed miRNAs in CRC patients and healthy controls, choosing 17 novel miRNAs with fold changes >5 for further validation. Plasma expression levels of miR-601 and miR-760 were significantly decreased in CRC compared to controls, showing at early stages a remarkedly decreased expression in respect to advanced adenomas and healthy controls. In addition, miR-601 and miR-760 expression was significantly reduced in advanced adenomas compared to healthy controls and increased compared to CRC. Interestingly, the combination of these biomarkers and CEA yielded an AUC of 0.805 and 80.4% sensitivity and 65.5% specificity, indicating their potential diagnostic value for early CRC [26]. Later, Kanaan et al. reported a panel of eight upregulated cell-free miRNAs (miR-532-3p, miR-331, miR-195, miR-17, miR-142-3p, miR-15b, miR-532 and miR-652) that differentiated advanced adenomas from healthy controls with 88% sensitivity and 64% specificity. Moreover, a 5-miRNA panel (miR-331, miR-15b, miR-21, miR-142-3p and miR-339-3p) distinguished patients with advanced adenomas from CRC with an AUC of 0.856 and 91% sensitivity and 69% specificity [29]. These findings suggest the existence of different miRNA profiles in the early steps of CRC carcinogenesis. In another study, miR-21, miR-29a and miR-125b showed significantly higher serum expression levels in patients with early colorectal neoplasia compared to healthy controls, showing AUC values of 0.706 for miR-21, 0.741 for miR-29a and 0.806 for miR-125b. Also, miR-21, miR-29a and miR-125b could discriminate patients with advanced neoplasia from non-advanced neoplasia and healthy controls showing AUC values that ranged from 0.690 to 0.731. Interestingly, the combination of these three miRNAs improved the discriminatory power for both early (AUC = 0.826) and advanced (AUC = 0.759) colorectal neoplasms. These three miRNAs were significantly increased in patients with tubular adenomas and high-grade intraepithelial neoplasms while miR-29a and miR-125b were also significantly elevated in tubule-villous adenomas and small colorectal neoplasms (≤5 mm) [27].

In addition to these studies, massive characterization of cell-free miRNAs between CRC patients and healthy individuals provided the researchers alternative miRNAs panels for early CRC detection. Thus, Hofsli et al. developed a model of 21 miRNAs that showed the same expression profile at early and advanced CRC stages. This serum-based miRNA model discriminated early CRC from healthy controls with 90% specificity and 87.5% sensitivity [30]. Wang et al. also identified a 3-miRNA panel (miR-7, miR-409-3p and miR-93) with high diagnostic accuracy for discriminating early CRC from healthy individuals (82% sensitivity and 89% specificity, AUC = 0.892) [28]. Using next generation sequencing, Vychytilova-Faltejskova et al. found 54 miRNAs significantly deregulated in the serum of 144 CRC patients and 96 healthy controls. Of these, 4 miRNAs (miR-23a-3p, miR-27a-3p, miR-142-5p and miR-376c-3p) were validated as diagnosis biomarkers demonstrating a high discriminatory power to detect early CRC (81% sensitivity and specificity, AUC = 0.877) [31]. More recently, a plasma 6-miRNA panel was validated for early CRC detection in a multicenter study including 96 CRC, 101 advanced adenomas and 100 healthy individuals. This miRNA signature demonstrated 95% sensitivity and 90% specificity for advanced adenomas and 94% sensitivity and 87% specificity for early CRC [32]. Furthermore, other miRNAs, such as miR-506, miR-4316, miR-182 and miR-30a-5p have also been evaluated as potential markers for early CRC detection [33,34,35].

Overall, all these studies combining different panels of cell-free miRNAs, reach high accuracy to detect the presence of CRC at early stages with rates of sensitivity and specificity above 80%, making evident the great potential of these molecules as screening biomarkers.

### 2.2. Cell-Free miRNAs for Advanced CRC Diagnosis

As previously mentioned, miRNAs have been involved in the advance of CRC, promoting invasion, migration and the progression of the disease through epithelial to mesenchymal transition (EMT) activation and other mechanisms behind the tumor spread [36]. Accordingly, researchers have examined the expression levels of circulating miRNAs in advanced stages and their diagnostic value for discriminating patients with lymph nodes and/or distant metastasis. In 2011, Cheng et al. analyzed 3 miRNAs (miR-21, miR-92a and miR-141) in the plasma of CRC patients, finding significantly higher levels of miR-141 in stage IV CRC patients compared to healthy individuals. Plasma miR-141 was significantly correlated with tumor stages and upregulated in distant compared to non-distant metastasis. Thus, ROC curve analysis showed that miR-141 discriminated stage IV from stages I–II with 66.7% sensitivity and 80.8% specificity, stage IV from stage III with 66.7% sensitivity and 89.7% specificity, and stage IV from stages I–III with 66.7% sensitivity and 84% specificity. In addition, the combination of miR-141 and CEA could detect additional metastatic CRC patients which were not detected using one of both biomarkers independently [37]. These results indicate that plasmatic miR-141 levels represent a non-invasive biomarker for detecting metastatic CRC patients. In this line, Wang and Gu evaluated the expression levels of miR-29a, miR-17-3p and miR-92a in serum of 38 metastatic CRC patients compared to 36 non-metastatic CRC cases. They found that serum miR-29a was significantly upregulated in patients with liver metastasis showing 75% sensitivity and specificity (AUC of 0.803) for discriminating metastatic from non-metastatic patients. They also observed a significant positive correlation between serum and tissue miR-29a expression levels in metastatic CRC patients, with its levels associated with advanced T stages [38]. Other studies have described upregulated levels of miR-126, miR-141, miR-21 and miR-200 in CRC patients with liver affectation [39,40]. Some of these miRNAs were associated with tumor metastasis through EMT regulation targeting ZEB1 and ZEB2 transcription factors [41]. Serum miR-885-5p high levels were also significantly associated with high TNM stage, lymph node and distant metastasis [42]. Recently, also upregulated serum levels of miR-103 were associated with lymph node metastasis and advanced tumor stage, reflecting its involvement in the progression of CRC [43]. Other circulating miRNAs expression patterns have been identified as potential biomarkers in CRC advanced stages. Giráldez et al. reported a plasma 3-miRNA panel (miR-19a, miR-19b and miR-15b) with high discriminatory power to detect advanced stages from healthy individuals with 76.19% sensitivity and 77.36% specificity (AUC = 0.81) [44]. In a similar study, Brunet-Vega et al. analyzed 11 upregulated tissue miRNAs in serum samples from 30 stage III CRC patients and 26 healthy individuals, finding only significantly increased expression of miR-18a and miR-29a in the patients’ cohort [45]. These results suggest the potential role of these miRNAs in the development of lymph node metastasis. Another miRNA analysis performed in 187 CRC patients and 47 healthy controls revealed significantly increased miR-141, let-7f-2, miR-628-5p, miR-203 and miR-200b expression levels in advanced stages compared to early stages and healthy individuals. In addition, miR-15b, miR-526, miR-96, miR-148a and miR-22 were also significantly upregulated in advanced stages compared to healthy individuals. Logistic regression analysis showed that plasma miR-203 differentiated advanced stages from early stages with 74.7% sensitivity and 71.4% specificity and, again, miR-141 distinguished stage IV from stages I–III with 80.0% sensitivity and 86.1% specificity [46]. Finally, downregulation of specific miRNAs was also described in advanced disease. Thus, serum levels of miR-139-3p were significantly downregulated in advanced stages CRC patients in respect to controls, showing 97.2% sensitivity and 97.8% specificity to identify the patients and demonstrating higher diagnostic accuracy compared to blood CEA levels [47].

### 2.3. Exosomal miRNAs for CRC Diagnosis

Nowadays, exosomal miRNAs have emerged as promising diagnostic biomarkers in cancer. Thus, Ogata-Kawata et al. identified 16 miRNAs significantly upregulated in serum exosomes from CRC patients compared to healthy controls as well as in colon cancer cell lines using a microarray-based approach. From those, serum exosomal let-7a, miR-1224-5p, miR-1229, miR-1246, miR-150, miR-21, miR-223 and miR-23a showed significantly lower levels after surgical resection of the primary tumor demonstrating the tumoral origin of these exosomal miRNAs. According to ROC curve analysis, miR-1246 and miR-23a showed sensitivities of 95.5% and 92% respectively, reflecting the high diagnostic power of both miRNAs. Notably, stage I patients showed significantly increased levels of seven miRNAs (let-7a, miR-1229, miR-1246, miR-150, miR-21, miR-223, and miR-23a), suggesting a role of exosomal miRNAs in the early steps of CRC development [48]. Wang et al. analyzed nine plasmatic exosomal miRNAs from 50 early CRC patients and matched healthy individuals but only miR-125-3p and miRNA-320c were significantly upregulated in CRC compared to healthy individuals, showing AUC values of 0.6849 and 0.5982, respectively. Interestingly, an important increase of diagnostic power for early CRC stage was observed by combining CEA and miR-125-3p (AUC = 0.8552) [49]. These data support evidence about the possibility to combine exosomal miRNAs and traditional tumor markers as useful CRC screening methods. MiR-21 has been also described as upregulated in plasmatic exosomes from CRC patients compared to healthy individuals. Moreover, miR-21 was higher in advanced respect to early stages and also in patients with liver metastasis [50]. Another circulating exosomal profile study including 77 CRC patients and 20 healthy individuals identified 7 deregulated miRNAs of which miR-638, miR-5787, miR-8075, miR-6869-5p and miR-548c-5p were significantly downregulated and miR-486-5p and miR-3180-5p were significantly upregulated. Among these, low expression of miR-638 was associated with a higher risk of liver metastasis and advanced TNM stage. Furthermore, a network analysis showed that miR-638, miR-5787, miR-8075, miR-6869-5p, and miR-548c-5p may regulate the glucose metabolism in CRC [51]. Recently, miR-17-5p, miR-181a-5p, miR-18a-5p and miR-18b-5p, which were detected previously as cell-free miRNAs in plasma, were described as upregulated in plasmatic exosomes. These findings suggest that circulating miRNAs are present in the bloodstream both bound to Ago2, as normally happens in cell-free miRNAs, or encapsulated in exosomes [52]. Also, exosomal miR-27a and miR-130a showed higher levels in plasma from CRC patients compared to healthy individuals and adenoma patients, providing the combination of both miRNAs a high diagnostic accuracy (AUC = 0.801) [53]. However, further research should be performed to reveal the mechanisms through which tumor cells secreted exosomes containing specific miRNAs in order to improve its clinical application as biomarkers and potential therapeutic targets.

## 3. miRNAs as Prognostic Biomarkers in CRC

The first report that established a relationship between circulating miRNAs and prognosis in CRC was performed in 2010 [54]. Since then, several promising circulating miRNAs were associated with poor progression-free survival (PFS) and overall survival (OS) times in CRC patients. Some of the miRNAs that have demonstrated diagnostic potential were also identified as useful prognostic biomarkers, while other miRNAs have been exclusively associated with CRC prognosis. Therefore, circulating miRNAs could be potential non-invasive biomarkers for CRC prognosis (Figure 3 and Appendix A).

### 3.1. Cell-Free miRNAs for CRC Prognosis

Circulating miRNAs that provide information about biological behavior and clinical evolution of CRC are good candidates for predicting disease prognosis. Pu et al. analyzed plasmatic levels of miR-21, miR-221 and miR-222 in 103 CRC patients and its correlation with p53, CEA, estrogen and progesterone receptors status. In this study, factors known as histological grade, clinical stage, presence of metastasis and high levels of miR-221 were significantly correlated with poor OS. Interestingly, a significant correlation between plasma miR-221 level and p53 protein expression was found but the role of miR-221 in the regulation of p53 is nowadays unknown [54]. Further, several works have found high levels of miR-17-3p [55], miR-106a [55], miR-21 [56], miR-92a [57], miR-1290 [58], miR-210 [59], miR-183 [60], miR-885-5p [42], miR-592 [61], miR-196b [62] or miR-155 [63] associated with lower PFS or OS. More recently, serum miR-203 has been proposed as another promising prognostic and metastasis-predictive biomarker in CRC. High circulating miR-203 levels were associated with high TNM stage, presence of lymph node infiltration, distant and peritoneal metastasis, and a shorter OS. ROC curve analysis showed that miR-203 could discriminate, with 62.5% sensitivity and 77.68% specificity, patients with poor prognosis. In addition, miR-203 was an independent biomarker for predicting the formation of lymph node, liver and peritoneal metastasis [64].

The prognostic impact of miRNA-200 family has been described in several studies focused on CRC [37,65,66,67]. Chen et al. showed that high plasma miR-141 levels were significantly associated with a poor OS in metastatic CRC patients [37]. Later, Toiyama et al. analyzed the expression levels of miR-200b, miR-200c, miR-141 and miR-429 in CRC serum samples, and they observed significantly elevated levels of miR-200c in stage IV tumors compared to stages I–III and healthy controls. Interestingly, high serum miR-200c levels were significantly associated with lymph node affectation and liver metastasis. Furthermore, multivariate logistic regression analysis showed that miR-200c was an independent prognostic marker for predicting shorter OS and tumor recurrence in stages II and III [67]. In another study, Maierthaler et al. performed a miRNA profiling in 20 metastatic and 20 non-metastatic patients, including in each group 10 patients with “good prognosis” and 10 patients with “bad prognosis”. A total of 11 miRNAs were validated in a large cohort of patients, finding significantly increased levels of miR-122, miR-141, miR-200a, miR-200b and miR-203a in metastatic compared to non-metastatic patients. According to univariate and multivariate analysis, “bad prognosis” patients showed significantly higher levels of miR-122 in the metastatic CRC group and in the non-metastatic CRC group significantly low levels of miR-200c were observed suggesting a different behavior of cell-free miRNAs during the tumor evolution. Further, high expression levels of miR-200a were only correlated with a shorter OS in metastatic CRC. Finally, high levels of miR-122 were associated with a shorter OS and relapse-free survival in both metastatic and non-metastatic CRC [65]. More recently, high levels of miR-200c and miR-141 in plasma obtained after mesenteric vein blood collection were associated with shorter OS in CRC. By contrary, these findings were not observed in blood from peripheral veins, which reflects a major potential for detecting tumor derived biomarkers from tumor-draining veins [68].

It is noteworthy to point out that downregulated miRNAs have also been associated with an unfavorable prognosis in CRC. Low levels of miR-30a-5p, miR-194, miR-29b and miR-23b were associated with a shorter PFS and OS [35,69,70]. These miRNAs were significantly associated with a high TNM stage. In particular, miR-23b was significantly associated with tumor depth, distant metastasis and tumor recurrence [70]. Currently, it is well known that the combination of various miRNAs with clinical data improves not only the ability to diagnose but also the prognosis. In fact, Vychytilova-Faltejskova et al. designed a signature for prediction of three-year OS consisting of three clinical factors (age, gender and stage) and two miRNAs (miR-23a-3p, miR-376c-3p). This model accurately predicted the outcome of more than 70% of CRC patients, even for each TNM stage independently [31]. Recently, Vafaee et al. developed a network based on an algorithm to discover robust miRNA signatures providing information about their biological role in CRC progression. By this approach, a plasma 11-miRNA signature was associated to several pathways related with CRC prognosis [71].

### 3.2. Exosomal miRNAs for CRC Prognosis

Several circulating exosomal miRNAs have been reported as potential prognosis biomarkers for CRC patients. Matsumura et al. identified six serum exosomal miRNAs associated with liver CRC metastasis after a global microarray profiling in 124 CRC patients. In this study, upregulated expression of miR-19a and miR-92a was observed in CRC patients compared to healthy controls, with high levels of exosomal miR-19a significantly associated with tumor infiltration, lymph node and liver metastasis and, therefore, with a higher risk of tumor recurrence and a worse overall prognosis [72]. Liu et al. validated as prognosis markers a total of 10 serum exosomal miRNAs identified by RNA sequencing in 84 samples of patients with stages II and III CRC collected after tumor resection and before adjuvant therapy. Downregulated levels of miR-4772-3p and upregulated levels of miR-4732-5p were observed in recurrent patients compared to the non-recurrent group. ROC curves demonstrated that miR-4772-3p allowed to discriminate recurrent from non-recurrent patients with 78.6% sensitivity and 77.1% specificity, which evidenced its potential as a prognostic biomarker for tumor recurrence [73]. Interestingly, Tsukamoto et al. evaluated the prognostic value of plasma exosomal miR-21 according to TNM classification. Among CRC patients, high miR-21 expression levels were significantly associated with a poor prognosis, determining a significantly worse disease-free survival (DFS) and OS in stages II–III CRC, while in stage IV only the OS was significantly worse in patients with high exosome-derived miR-21. Importantly, the multivariate analysis revealed that miR-21 was an independent prognostic biomarker for DFS and OS in stages II–III and for OS in stage IV [50]. Similarly, high serum exosomal miR-6803-5p levels were significantly and independently associated with shorter DFS and OS in CRC patients, with this predictive value higher in patients with advanced stages and liver metastasis [74]. Other exosome-encapsulated miRNAs associated with a poor prognosis in CRC were miR-548c-5p, miR-27a and miR-130a [53,75]. In particular, decreased expression levels of miR-548c-5p were significantly associated with a shorter OS, with this downregulation more significant in patients with liver metastasis and advanced tumor stages. Then, multivariate analysis demonstrated that low levels of miR-548c-5p were independently associated with poor OS. Interestingly, potential miR-548c-5p target genes, such as STAT3, PTPRO, IRF, interleukins and their receptors, implicated in CRC carcinogenesis and dissemination have been described providing valuable information regarding the role of this miRNA in the context of CRC [75]. Recently, Fu et al. carried out an exosomal miRNA sequencing from primary (SW480) and lymph node metastatic (SW620) CRC cell lines. Expression levels of 11 deregulated miRNAs (miR-17, miR-19a, miR-20, miR-92a, miR-7, miR-181a, miR-375, miR-194, miR-30d, miR-192, and miR-146) were further validated in serum exosomes from 29 CRC patients and 11 healthy individuals, finding significantly increased levels of miR-17 and miR-92a in metastatic CRC compared to controls. In addition, circulating exosomal levels of both miRNAs were significantly correlated with pathological stages and grades of CRC, suggesting their potential as prognostic biomarkers for primary and metastatic CRC [76].

## 4. miRNAs as Therapy Prediction Biomarkers in CRC

Chen et al. were the first researchers that explored the role of circulating miRNAs as non-invasive biomarkers for predicting the development of resistance to chemotherapy in CRC [77]. Due to cancer patients frequently showing resistance to therapy during the course of the disease, there is an urgent need to find precise predictive biomarkers that choose effective drugs for each patient and identify early therapy resistance to avoid overtreatments with the consequent toxic effects.

### 4.1. Cell-Free miRNAs for CRC Therapy Prediction

Over the last few years, several studies [77,78,79] have focused on plasma and serum miRNAs as potential biomarkers to predict the sensitivity of CRC to chemotherapy. Chen et al. characterized serum samples from 16 responders and non-responders CRC patients under FOLFOX based chemotherapy using microarrays. They found five differentially expressed miRNAs (miR-221, miR-222, miR-122, miR-19a, miR-14) that were further validated in 72 CRC patients. From them, only serum miR-19a was significantly upregulated in non-responder patients, being able to discriminate non-responder from responder patients with 66.7% sensitivity and 63.9% specificity [77]. Similarly, Zhang et al. performed a genome-wide expression profiling to identify predictive serum miRNAs for oxaliplatin-based chemotherapy. After this global gene expression analysis, the authors found five miRNAs (miR-20a, miR-130, miR-145, miR-216 and miR-372) significantly downregulated in a large-scale validation phase, including 93 responder and 80 non-responder patients. This 5-miRNA panel showed 92% sensitivity and 88% specificity to discriminate primary sensitive and resistant patients, demonstrating the potential of this panel as a predictive tool for guiding CRC therapy selection [78]. In another study, Kjersem et al. evaluated the expression of 742 plasma miRNAs in metastatic CRC patients before and after chemotherapy. Before onset treatment, the upregulation of three miRNAs (miR-106a, miR-130b and miR-484) was associated with a lack of response to 5-fluorouracil and oxaliplatin but not with a reduced survival. In addition, high plasma levels of miR-148 and miR-27b were associated with a shorter PFS and high expression of miR-326 was associated with shorter PFS and OS [79]. Recently, Ji et al. discovered a serum 4-miRNA signature (miR-328-3p, miR-652-3p, miR-342-3p and miR-501-3p) significantly correlated with therapeutic outcome in stage II–III CRC patients who were treated with fluorouracil-based adjuvant chemotherapy with or without leucovorin, levamisole or oxaliplatin. In both stages, a high-risk signature score was significantly associated with a poor DFS and OS, indicating the potential of this miRNA signature as an independent predictor of the response to adjuvant chemotherapy [80].

Interestingly, miRNAs have been also involved in angiogenic signaling and vascular integrity [81]. Hansen et al. analyzed the predictive value of miR-126 in 68 CRC patients treated with first-line chemotherapy combined with bevacizumab. During treatment, upregulated levels of plasma miR-126 were related with a lack of response to the anti-angiogenic therapy. Also, changes in tumor size were significantly correlated with the plasmatic levels of miR-126. Further, survival analysis revealed favorable prognosis in those patients with high miR-126 levels at baseline and a decrease of these levels during the treatment [82]. Similarly, Ulivi et al. analyzed a panel of plasmatic miRNAs involved in the angiogenic process with value as biomarkers to predict bevacizumab efficacy in advanced CRC. High baseline levels of miR-20b-5p, miR-29b-3p and miR-155-5p were significantly associated with better PFS and OS rates. Importantly, an increase of ≥30% of miR-155-5p after one month of therapy was significantly associated with shorter PFS and OS, indicating its utility for the treatment monitoring. The authors of this study proposed these three miRNAs as potentially involved in the angiogenic regulation, acting as predictive markers to select bevacizumab treatment in metastatic CRC patients [83]. In addition, high circulating miR-345 levels were associated with a lack of response to cetuximab and irinotecan with an odds ratio of 5.37 between the miR-345 high- and low-expression group [84].

Up to date, neoadjuvant chemoradiotherapy is considered the primary treatment in patients with locally rectal advanced cancer. However, preoperative chemoradiotherapy has not been associated with an OS benefit, which indicates the need for new predictive biomarkers that identify those CRC patients who benefit from this adjuvant therapy [85]. Based on a microarray approach, Yu et al. analyzed the circulating miRNA profile in patients with locally advanced rectal cancer that underwent preoperative chemoradiotherapy (before and after therapy). In this study, the authors found that serum miR-345 expression was significantly downregulated in chemoradiotherapy sensitive patients compared to the chemoradiotherapy-resistant group, showing an AUC of 0.75. According to the survival analysis, low miR-345 expression was associated with superior three-year local recurrence-free survival [86]. In addition, serum miR-125b has also been reported as another predictive biomarker of the preoperative chemoradiotherapy responsiveness in patients with rectal adenocarcinoma. Thus, high serum miR-125b levels identified non-responder from responder patients (AUC = 0.782), observing the same pattern in CRC tissue [87].

### 4.2. Exosomal miRNAs for CRC Therapy Prediction

Only two studies have described the potential predictive value of exosomal miRNAs for evaluating the therapeutic efficacy in CRC [73,88]. In this study tumor recurrence after FOLFOX treatment in stage III CRC patients was detected in 90% of the cases with low miR-4772-3p expression levels compared to patients with high expression levels. Although the authors consider that these results need to be validated in a large cohort of patients, their study reveals the potential predictive value of exosomal miRNAs for response to adjuvant chemotherapy. Recently, Jin et al. discovered a panel of 4-serum exosomal miRNAs (miR-21-5p, miR-1246, miR-1229-5p and miR-96-5p) which could discriminate resistant from sensitive CRC patients to conventional chemotherapy with 78% sensitivity and 88.90% specificity [88]. These findings provide evidence about the potential of exosomal miRNAs for predicting the resistance to chemotherapy in CRC.

## 5. Conclusions and Future Perspectives

Our review highlighted the potential of circulating miRNAs as novel biomarkers for early detection, diagnosis, prognosis and predictive therapy of CRC. Despite the extensive research in this field, there are still several challenges that remain to be overcome before their clinical application. Several studies have identified associations between circulating miRNA expression patterns and the diagnosis, prognosis and sensitivity to chemotherapy in CRC; however, a further investigation regarding the origin of miRNAs and their biological function is needed. As regulators of gene expression, numerous studies have demonstrated the potential of circulating miRNAs as predictive therapy biomarkers in CRC, however, to elucidate the mechanisms through which miRNAs are involved in the resistance to chemotherapy and other targeted therapies are required. miRNA-based anticancer therapies are being developed, alone or also in combination with other therapies, with the clear objective of overcoming the development of therapy resistance. The option to monitor the status of these miRNAs in liquid biopsies will be a key element in order to apply the anti-miRNA therapeutic strategies during the CRC evolution. Importantly, the mechanisms behind miRNAs alteration in CRC are still quite unknown. To improve the knowledge about these mechanisms would be of great value to design the most accurate miRNAs panel for monitoring CRC patients at different stages of the disease.

In fact, although numerous miRNAs have been examined with the same clinical purpose, there is not a coincidence between single miRNAs or miRNA panels reported by many studies, and even between studies that analyzed the same miRNAs there are sometimes opposite results. These heterogeneous results could be explained by the selection strategy of patients (sample sizes, tumor stages, socio-demographic and clinicopathological characteristics), the samples collection (type of body fluid, collection tubes and protocols), the sample processing and the different approaches for the miRNAs analysis, including data normalization (endogenous or exogenous normalizers) and the use of different cutoff values for the same miRNAs. For example, the harmonization of circulating exosome extraction constitutes a key element to reach a higher reproducibility in studies characterizing exosome-derived miRNAs. Therefore, consensus of the scientific community should be carried out to standardize pre-analytical and post-analytical protocols and to determine the best housekeeping genes to establish a gold standard normalization method. Several studies have demonstrated that cancer-associated circulating miRNAs can be detected in many biofluids, such as urine or saliva. Further studies comparing miRNA profiles in blood and other fluids of the same patient are required to determine the best sample source for a clinical test in CRC patients. In addition, prospective studies including large scale cohorts from multiple centers represent an urgent need to reach an effective clinical implementation of circulating miRNAs-based tests to guide CRC management.

Nowadays, it seems unlikely to use a single miRNA as a diagnostic, prognostic or predictive biomarker for CRC. Several miRNAs have been deregulated in different malignancies, which suggests that miRNAs could not be specific of a cancer type. In this sense, the combination of multiple miRNAs or even the combination of miRNA panels with traditional biomarkers such as CEA or CA19-9 or with clinical variables may provide an increase of the diagnostic accuracy for CRC. Interestingly, miRNA panels are based on the combination from two miRNAs up to several miRNAs. However, an ideal clinical model must have both a high sensitivity and specificity and a suitable cost-effectiveness. Although several miRNA-panels for CRC have been reported, until now, cost–benefit analysis has not yet been evaluated. It is necessary to developed cost–benefit models that determine the number of miRNAs that should be included, considering the possible combination with other biomarkers. This approach will allow researchers to know the real possibilities to implement the use of these biomarkers from prevention to treatment management. Thus, this promising strategy will apply a more personalized medicine in CRC.

In conclusion, although future studies must address some important challenges in this field, actually circulating miRNAs could be considered as a molecular mirror of the underlying changes that occur in the development and progression of CRC, providing valuable information on early and advanced diagnosis, prognosis, response to therapy and prediction of treatment response.

## Figures and Tables

**Figure 1 cancers-11-00898-f001:**
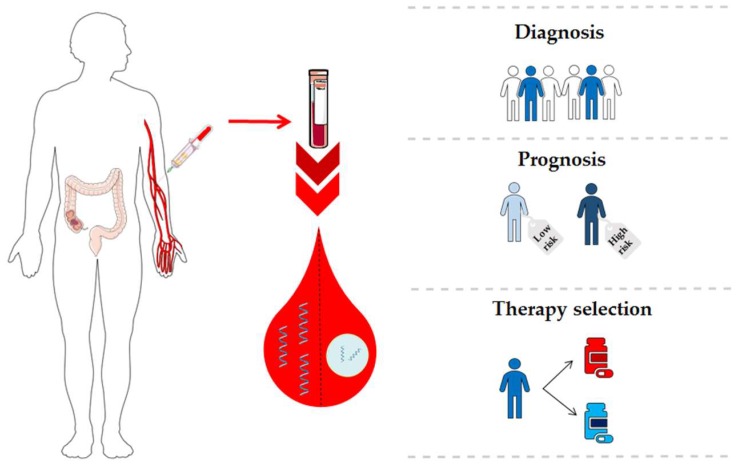
Clinical applications of circulating microRNAs (miRNAs) in colorectal cancer (CRC).

**Figure 2 cancers-11-00898-f002:**
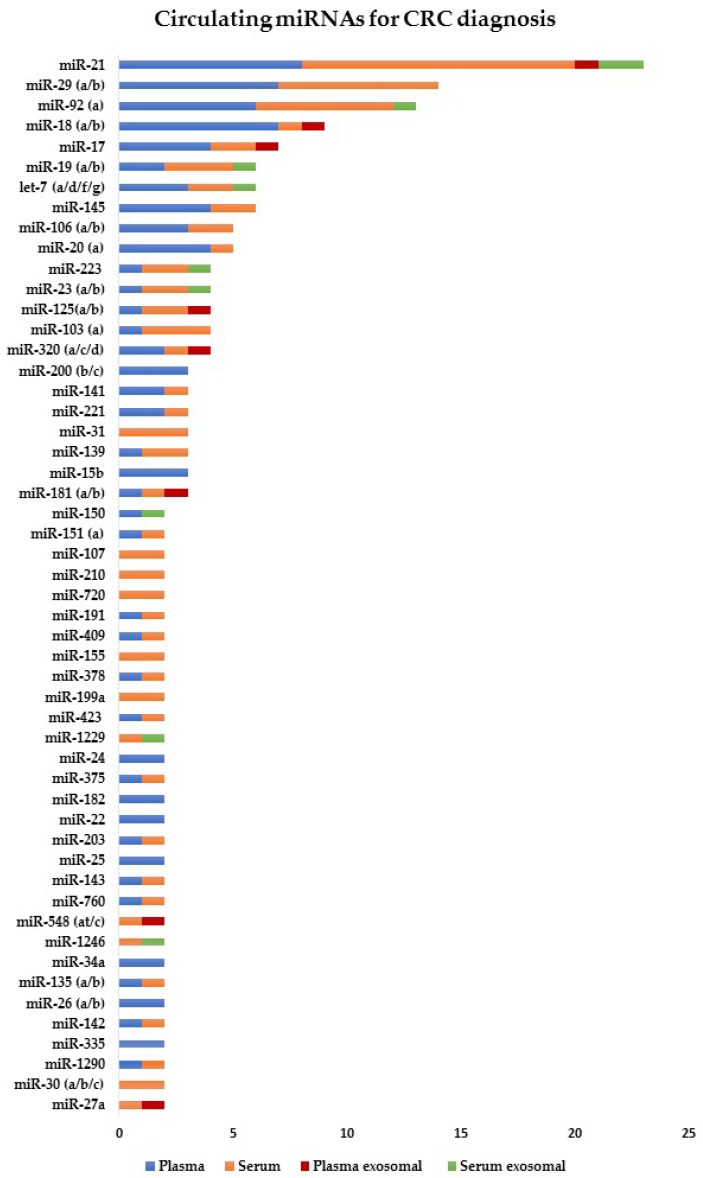
List of circulating miRNAs reported in two or more studies for CRC diagnosis. These miRNAs were significantly deregulated in CRC compared to healthy individuals (*p* < 0.05). References are provided in Appendix A.

**Figure 3 cancers-11-00898-f003:**
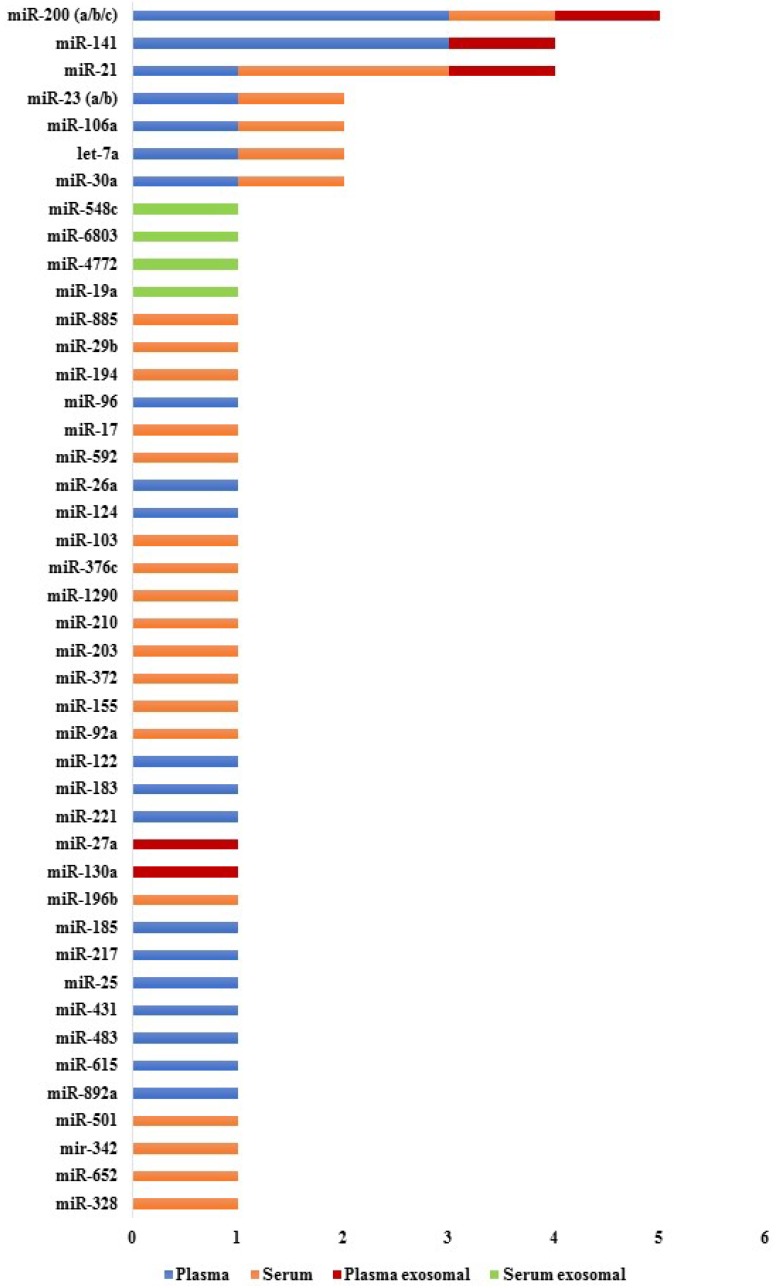
List of circulating miRNAs reported in one or more studies for CRC prognosis. These miRNAs were significantly associated with survival (*p* < 0.05). References are provided in Appendix A.

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
