# Peer review of "Circulating microRNAs as Promising Biomarkers in Colorectal Cancer"

_cancers, 2019, doi:10.3390/cancers11070898_

Round 1
Reviewer 1 Report
The manuscript entitled “Circulating microRNAs as Promising Biomarkers in Colorectal Cancer” by Rapado-Gonzalez et al. reviews the determination of miRNAs by liquid biopsies as a tool to expand the actual repertory of biomarkers in CRC. The field of miRNAs in CRC is an interesting one and, despite having been often reviewed, the present manuscript exhibits some features which are entirely its own. After a brief introduction, the authors deal with the use of miRNAs in diagnosis, prognosis and therapy selection in CRC. The senior author has published several papers on liquid biopsy in a number of cancer types and, therefore, she has a first-hand expertise in the subject.
Nevertheless, the organisation of the manuscript makes it difficult to read. Under every heading, the authors list a series of miRNAs over- or underexpressed when CRC patients are compared with healthy individuals. They add some details, such as the sensitivity and specificity of the analyses and, in some instances, the corresponding values for miRNAs panels were given. The abundance of data and their presentation as a continuous written account may conceal some of the otherwise valuable data. I suggest the authors may present their data in the form of tables, giving in different columns the name of the miRNA and all the items they want to include when available (e. g., cohort size, up- or down-regulated, sensitivity and specificity, CRC type and/or staging, source of sample, analytical methods, pathways affected by the miRNA, reference, and so on). The potential reader would get in this way a rapid access to the data. The authors may follow the model used in many other reviews on similar subjects, some of them in Cancers. Written text may be used to emphasize the main conclusions of the recorded data. The figures especially figs. 2 and 3, as they stand, are scarcely informative.
Apart from this main objection, some other minor details ought to be taken into account.
1) The reference given on line 69 is perhaps not adequate to illustrate the origin and role of miRNAs, as it is too specific for CRC. The article by Shirafkan et al., dealing with the use of miRNAs as biomarkers in cancer, may be quoted in other context of the present manuscript, bu here a more general review on the nature and role of miRNAs should be given. There are many excellent reviews in this sense.
2) The sentence starting “It is also widely accepted…” (lines 72-73) must be corrected, as its actual wording may suggest that miRNAs are genes and not their products.
3) The sentence “Therefore, the expression…” (lines 86-87) is true, but only when the changes in miRNA expression occur at an early stage of oncogenesis.
4) As aptly mentioned, Hofsli et al. developed a model including 21 miRNAs to discriminate the early onset of CRC, but this was based on the fact that the expression of these miRNAs was concordant between stage IV and stages I and II. If this circumstance is not mentioned, it is hard understanding how a miRNA profile of stage IV patients may lead to an early detection of CRC.
Author Response
Response to Reviewer 1 Comments
Point 1: The abundance of data and their presentation as a continuous written account may conceal some of the otherwise valuable data. I suggest the authors may present their data in the form of tables, giving in different columns the name of the miRNA and all the items they want to include when available (e. g., cohort size, up- or down-regulated, sensitivity and specificity, CRC type and/or staging, source of sample, analytical methods, pathways affected by the miRNA, reference, and so on). The potential reader would get in this way a rapid access to the data. The authors may follow the model used in many other reviews on similar subjects, some of them in Cancers. Written text may be used to emphasize the main conclusions of the recorded data. The figures especially figs. 2 and 3, as they stand, are scarcely informative. Apart from this main objection, some other minor details ought to be taken into account.
Response 1: We agree with the reviewer regarding the need of additional tables to facilitate the comprehension of so much information. Therefore, following the reviewer instructions we have included a table with the following information: authors, study cohort, TNM stage, technique, specimen, molecular profile, sensitivity, specificity, expression level, normalization and clinical application. It is Supplementary table number 1.
This table resume all the information of each study.
Point 2: The reference given on line 69 is perhaps not adequate to illustrate the origin and role of miRNAs, as it is too specific for CRC. The article by Shirafkan et al., dealing with the use of miRNAs as biomarkers in cancer, may be quoted in other context of the present manuscript, but here a more general review on the nature and role of miRNAs should be given. There are many excellent reviews in this sense.
Response 2: Taking into account the suggestion of the reviewer we have removed the article by Shirafkan and we have included a more adequate reference. This is the following: Lin, S.; Gregory, R.I. MicroRNA biogenesis pathways in cancer. Nat. Rev. Cancer. 2015, 15, 321-33.
Point 3: The sentence starting “It is also widely accepted…” (lines 72-73) must be corrected, as its actual wording may suggest that miRNAs are genes and not their products.
Response 3: Following the reviewer indication this sentence was corrected in the new version of the manuscript.
Line 72-75: “It is also widely accepted that upregulation of miRNAs entails oncogenesis by downregulating tumor suppressor genes [14]. They modulate specific individual mRNA targets or multiple mRNAs [15] and show aberrant expression patterns in different tumor types [16-18]”.
Point 4: The sentence “Therefore, the expression…” (lines 86-87) is true, but only when the changes in miRNA expression occur at an early stage of oncogenesis.
Response 4: Following the reviewer instructions, we have removed the sentence “Therefore, the expression profile of circulating miRNAs may reflect the miRNA profile of tumor tissue, facilitating the early detection of tumor in a non-invasive manner.” and we have clarified the information with a new sentence (line 86-88) “Therefore, the expression profile of circulating miRNAs may reflect the miRNA profile of primary tumor or metastatic lesions, facilitating the detection of the tumor at different stages in a non-invasive manner”.
Point 5: As aptly mentioned, Hofsli et al. developed a model including 21 miRNAs to discriminate the early onset of CRC, but this was based on the fact that the expression of these miRNAs was concordant between stage IV and stages I and II. If this circumstance is not mentioned, it is hard understanding how a miRNA profile of stage IV patients may lead to an early detection of CRC
Response 5: Now, we have clarified the information of the sentence. We have removed “Thus, Hofsli et al. developed a model of 21 miRNAs based on miRNA profile of stage IV CRC patients that allowed to discriminate early CRC from healthy controls with 90% specificity and 87.5% sensitivity [30]” and we have included (line 142-144) “Thus, Hofsli et al. developed a model of 21 miRNAs that showed the same expression profile at early and advanced CRC stages. This serum-based miRNA model allowed to discriminate early CRC from healthy controls with 90% specificity and 87.5% sensitivity [30]”.
Reviewer 2 Report
The review article entitled “Circulating microRNAs as Promising Biomarkers in Colorectal Cancer” by Oscar Rapado- Gonzalez et al focused on the clinical potential of blood circulating miRNAs as emerging biomarkers with high value for improving the clinical management for colorectal cancer (CRC) patients.
This review is a thorough study and it is of utmost interdisciplinary interest and importance. The author significantly described the impact of miRNA in the initial and adverse stage of colon cancer in different sections, such as cell-free miRNA in early and advanced CRC diagnosis, exosomal miRNAs for CRC diagnosis, miRNAs a prognostic biomarkers and therapy prediction in CRC and so on.
A limited study is found in CRC therapy prediction by exosomal miRNAs.
However, a panel of miRNAs either cell-free or exosomal are involved in most cases for CRC diagnosis, prognosis, or therapy prediction.
The reviewer is wondering how it could be cost effective for the selection of miRNA as a CRC biomarker.
Author Response
Response to Reviewer 2 Comments
Point 1: The review article entitled “Circulating microRNAs as Promising Biomarkers in Colorectal Cancer” by Oscar Rapado- Gonzalez et al focused on the clinical potential of blood circulating miRNAs as emerging biomarkers with high value for improving the clinical management for colorectal cancer (CRC) patients. This review is a thorough study and it is of utmost interdisciplinary interest and importance. The author significantly described the impact of miRNA in the initial and adverse stage of colon cancer in different sections, such as cell-free miRNA in early and advanced CRC diagnosis, exosomal miRNAs for CRC diagnosis, miRNAs a prognostic biomarkers and therapy prediction in CRC and so on.
A limited study is found in CRC therapy prediction by exosomal miRNAs. However, a panel of miRNAs either cell-free or exosomal are involved in most cases for CRC diagnosis, prognosis, or therapy prediction.
Response 1: Several panels have been described in our manuscript for CRC diagnosis and prognosis. Regarding CRC therapy prediction by exosomal miRNAs, we have reviewed the scientific literature, once again, and we have found another recent reference. We have included the following information in section 2.3 called: “Exosomal miRNAs for CRC therapy prediction”:
Line 404-408 “Recently, Jin et al. discovered a panel of 4-serum exosomal miRNAs (miR-21-5p, miR-1246, miR-1229-5p and miR-96-5p) which could discriminate resistant from sensitive CRC patients to conventional chemotherapy with 78% sensitivity and 88.90 % specificity [88]. These findings provide evidence about the potential of exosomal miRNAs for predicting the resistance to chemotherapy in CRC.”
Point 2: The reviewer is wondering how it could be cost effective for the selection of miRNA as a CRC biomarker.
Response 2: We appreciate the comment of the review because we didn´t include this issue in the first version because of the absence of clear data about this matter, however, it is a key point for the clinical translation of these biomarkers. Therefore, we have included the following sentence regarding the cost-effective selection of miRNA as a CRC.
Line 451-456: “Although several miRNA-panels for CRC have been reported, until now, cost-benefit analysis has not yet been evaluated. It is necessary to developed cost-benefit models that determine the number of miRNAs that should be included, taking into account the possible combination with other biomarkers. This approach will allow to know the real possibilities to implement the use of these biomarkers from prevention to treatment management. Thus, this promising strategy will allow to apply a more personalized medicine in CRC.”
Reviewer 3 Report
In the manuscript titled, “Circulating microRNAs as Promising Biomarkers in Colorectal Cancer”, the authors provided a comprehensive review of the circulating microRNAs in colorectal cancer and highlighted their potential as emerging biomarkers in early detection, prognosis, and therapy of colorectal cancer. The manuscript is well-written, easy to follow, and would be interesting to the target audience. The manuscript would be an important addition to the journal.
The authors are recommended to add references of the articles from where the figures are generated either in the corresponding figure or sub-section of the manuscript.
Author Response
Response to Reviewer 3 Comments
Point 1: In the manuscript titled, “Circulating microRNAs as Promising Biomarkers in Colorectal Cancer”, the authors provided a comprehensive review of the circulating microRNAs in colorectal cancer and highlighted their potential as emerging biomarkers in early detection, prognosis, and therapy of colorectal cancer. The manuscript is well-written, easy to follow, and would be interesting to the target audience. The manuscript would be an important addition to the journal.
The authors are recommended to add references of the articles from where the figures are generated either in the corresponding figure or sub-section of the manuscript.
Response 1: Following the reviewer suggestions, we have included two tables as supplementary fields adding references of the articles from the figures 2 and 3. These are the following:
Supplementary Table 2. List of references of Figure 2.
miRNAs | References | miRNAs | References |
miR-21 | [1–22] | miR-720 | [15,32] |
miR-29 (a/b) | [5,18,19,23–33] | miR-191 | [32,34] |
miR-92 (a) | [6,7,18,19,23,27,31,32,34–37] | miR-409 | [32,64] |
miR-18 (a/b) | [9,18,24,25,33,34,38,39] | miR-155 | [32,65] |
miR-17 | [10,27,36,39–41] | miR-378 | [8,32] |
miR-19 (a/b) | [21,24,33,37,40,41] | miR-199a | [32,66] |
let-7 (a/d/f/g) | [7,12,13,32,42,43] | miR-423 | [32,52] |
miR-145 | [10,12,26,31,40] | miR-1229 | [13,22] |
miR-106 (a/b) | [18,32,40,44,45] | miR-24 | [34,52] |
miR-20 (a) | [12,18,41,45,46] | miR-375 | [35,67] |
miR-223 | [13,19,34,37,41] | miR-182 | [46,68] |
miR-23 (a/b) | [13,47–49] | miR-22 | [9,42] |
miR-125 (a/b) | [5,19,27,50] | miR-203 | [7,42] |
miR-103 (a) | [15,32,39,51] | miR-25 | [9,64] |
miR-320 (a/c/d) | [32,50,52,53] | miR-143 | [18,32] |
miR-200 (b/c) | [12,38,42,54,55] | miR-760 | [35,69] |
miR-141 | [42,54,56,57] | miR-548 (c/at) | [59,70] |
miR-221 | [32,34,58] | miR-1246 | [13,22] |
miR-31 | [7,12,57] | miR-34a | [32,62,71] |
miR-139 | [59–61] | miR-135 (a/b) | [12,72] |
miR-15b | [10,24,33] | miR-26 (a/b) | [10,73] |
miR-181 (a/b) | [7,39] | miR-142 | [48,73] |
miR-150 | [13,62] | miR-335 | [24,33] |
miR-151 (a) | [32,39] | miR-1290 | [53,74] |
miR-107 | [32,59] | miR-30 (a/b/c) | [75,76] |
miR-210 | [32,63] | miR-27a | [48,77] |
Supplementary Table 3. List of references of Figure 3.
miRNAs | References | miRNAs | References |
miR-200 (a/b/c) | [1–4] | miR-210 | [25] |
miR-141 | [5,1,4] | miR-203 | [26] |
miR-21 | [6–9] | miR-372 | [10] |
miR-23 (a/b) | [10,11] | miR-155 | [27] |
miR-106a | [8,12] | miR-92a | [28] |
let-7a | [8,13] | miR-122 | [2] |
miR-30a | [8,14] | miR-183 | [29] |
miR-548c | [15] | miR-221 | [30] |
miR-6803 | [16] | miR-27a | [31] |
miR-4772 | [17] | miR-130a | [31] |
miR-19a | [18] | miR-196b | [32] |
miR-885 | [19] | miR-185 | [8] |
miR-29b | [20] | miR-217 | [8] |
miR-194 | [20] | miR-25 | [8] |
miR-96 | [1] | miR-431 | [8] |
miR-17 | [12] | miR-483 | [8] |
miR-592 | [21] | miR-615 | [8] |
miR-26a | [22] | miR-892a | [8] |
miR-124 | [22] | miR-501 | [33] |
miR-103 | [23] | miR-342 | [33] |
miR-376c | [10] | miR-652 | [33] |
miR-1290 | [24] | miR-328 | [33] |

Round 2
Reviewer 1 Report
The revised version of the manuscript of ref. cancers 517108 incorporates most of the suggestions done in the first review and the authors have provided satisfactory answers to the queries posed. They have incorporated, as supplementary material, a table summarizing the findings on circulating miRNAs and this will make the reported information easier to read. No other objections are raised.